# Anogenital HPV-Related Cancers in Women: Investigating Trends and Sociodemographic Risk Factors

**DOI:** 10.3390/cancers16122177

**Published:** 2024-06-08

**Authors:** Micol Lupi, Sofia Tsokani, Ann-Marie Howell, Mosab Ahmed, Danielle Brogden, Paris Tekkis, Christos Kontovounisios, Sarah Mills

**Affiliations:** 1Department of Surgery and Cancer, South Kensington Campus, Imperial College London, London SW7 2AZ, UK; danielle.brogden@nhs.net (D.B.); p.tekkis@imperial.ac.uk (P.T.); c.kontovounisios@imperial.ac.uk (C.K.); sarah.mills58@nhs.net (S.M.); 2Department of Colorectal Surgery, Chelsea and Westminster NHS Foundation Trust, 369 Fulham Road, London SW10 9NH, UK; ann-marie.howell2@nhs.net; 3Laboratory of Hygiene, Social & Preventive Medicine and Medical Statistics, School of Medicine, Aristotle University of Thessaloniki, 541 24 Thessaloniki, Greece; stsokani@auth.gr; 4Cochrane Methods Support Unit, Evidence Production and Methods Department, Cochrane, London W1G 0AN, UK; 5Department of Anesthesiology, State University of New York Downstate Health Sciences University, 450 Clarkson Avenue, Brooklyn, NY 11203, USA; 6Department of Colorectal Surgery and Cancer, The Royal Marsden NHS Foundation Trust, 203 Fulham Road, London SW3 6JJ, UK; 7Evangelismos General Hospital, Ipsilantou 45-47, 106 76 Athens, Greece

**Keywords:** HPV, SCC, genital, anal, women

## Abstract

**Simple Summary:**

Anal cancer incidence is rising, especially in women. This is an issue because anal cancer has a precancerous stage, which can be identified and treated before it progresses to cancer. Nevertheless, not all women would benefit from anal cancer screening and precancer detection. Given that women with other anogenital cancers are likely to share similar risk factor profiles, this study aims to explore the incidence trends of all anogenital cancers in women in England between 2014 and 2020, as well as the role of important factors such as age, deprivation and ethnicity in these incidences. The findings from this study will highlight the populations at highest risk of these cancers and will aid the implementation of an anogenital cancer prevention strategy.

**Abstract:**

The incidences of anogenital HPV-related cancers in women are on the rise; this is especially true for anal cancer. Medical societies are now beginning to recommend anal cancer screening in certain high-risk populations, including high-risk women with a history of genital dysplasia. The aim of this study is to investigate national anogenital HPV cancer trends as well as the role of demographics, deprivation, and ethnicity on anogenital cancer incidence in England, in an attempt to better understand this cohort of women which is increasingly affected by anogenital HPV-related disease. Demographic data from the Clinical Outcomes and Services Dataset (COSD) were extracted for all patients diagnosed with anal, cervical, vulval and vaginal cancer in England between 2014 and 2020. Outcomes included age, ethnicity, deprivation status and staging. An age over 55 years, non-white ethnicity and high deprivation are significant risk factors for late cancer staging, as per logistic regression. In 2019, the incidences of anal and vulval cancer in white women aged 55–74 years surpassed that of cervical cancer. More needs to be done to educate women on HPV-related disease and their lifetime risk of these conditions.

## 1. Introduction

The Human Papillomavirus (hrHPV) is responsible for approximately 4.8% of all cancers worldwide [1]. It is a non-enveloped double-stranded DNA virus with over 400 known genotypes [2]. A total of 30 of these genotypes are sexually transmissible and several have been shown to be carcinogenic to the anogenital region; typically, high-risk HPV (hrHPV) types 16 and 18 [3]. Most sexually active women and men will be infected with HPV in their lifetime, regardless of the their sexual practices, and up to 50% of hrHPV infections will clear within 6 months to 2 years after acquisition, with only 10% of persistent infections eventually causing dysplasia [4]. hrHPV is responsible for 40% of vulval cancers, 70% of vaginal cancers, 90% of cervical cancers and 90% of anal cancers [1,5,6].

The introduction of the national cervical cancer screening programme in England in 1988 has decreased the incidence of cervical cancer by over a third [7]; however, the incidence of other anogenital cancers in women has been on the rise, and this is especially true for anal, but also vulval, squamous cell carcinoma (SCC) [8,9,10]. Given that hrHPV exposure is most common in sexually active women under 30 years of age [11,12] and that cervical high-grade squamous intraepithelial lesions (HSIL) and cancer tend to present in women under the age of 50 [12], whilst vulval and anal cancer tend to present in those above the age of 50 years [13,14], it appears that women are being treated for their cervical pathology only to be later affected by vulval and anal pathologies, for which no national screening programmes exist.

A persistent cervical hrHPV infection, HSIL and/or cancer could therefore in theory act as a potential early determinant of latent vulval and/or anal disease. There are now numerous studies describing these relationships between cervical and other anogenital pathologies; Papatia et al. [15] found that cervical cancer patients aged between 20 and 53 years have an increased risk of developing anal cancer (SIR 3.53, 95% CI 1.15–8.23), a risk which was significant 10 or more years after the cervical cancer diagnosis and not seen in those patients diagnosed with cervical cancer after the age of 53 years. Kalliala et al. [16] reports the relative risk of vaginal (10.84, CI 5.58–21.10, *p* < 0.001), vulval (3.34, CI 2.39–4.67, *p* < 0.001) and anal cancer (5.11, CI 2.73–9.55, *p* < 0.001) to be higher in women previously treated for cervical HSIL compared to the general population. A systematic review by Clifford et al. [17], looking at anal SCC incidence rates in high risk populations, found elevated anal cancer incidence rates of 9, 6, 48, 42, 10, 19 per 100,000 people/year for women with cervical cancer, cervical precancerous lesions, vulval cancer, vulval precancerous lesions, vaginal cancer and vaginal precancerous lesions, respectively (compared to the baseline incidence of 1–2 per 100,000 people/year [18,19]).

There is clearly a large amount of overlap between different hrHPV-driven anogenital pathologies, which could be arguably considered one entity, given that at least 30% of women with cervical and/or vulval HSIL have been shown to have anal HSIL [4,20,21,22], a percentage which has been shown to increase to 57% in women with more than one focus of HSIL [21]. With a 10- to 20-year lag between cervical and anal disease, as well as limited research, lack of anal disease surveillance and the fact that cervical screening has only recently changed in 2016 from cytology to hrHPV testing, it is possible that women affected by vulval, and anal disease were colonised with hrHPV in the cervix earlier in life. As a matter of fact, a study looking at women with multizonal anogenital disease found the cervix to be the most affected initial site of anogenital disease. Perianal and anal canal pathologies, on the other hand, were most commonly seen as secondary new sites of disease later in life [23]. In concordance with these findings, a study by Wei et al. [24] showed age-specific shifts in HPV-16 prevalence from cervix to anus, suggesting that anal infections persist longer or occur later in life than the cervix.

Nevertheless, the profile of women at high-risk of anal cancer is still poorly understood, namely due to the fact that this group of patients is poorly researched, with most of anal cancer research focusing on high-risk men and in people living with HIV (PLWH).

This is despite the fact that two thirds of all anal cancer diagnoses are in women [25] and that only 3% of the global female anal SCC burden is in PLWH [26]. Smoking, immunosuppression, number of sexual partners and HIV are all significant risk factors for HPV-related disease [27], However, they do not explain the entire picture.

With the recent publication of the ANCHOR trial, demonstrating that the treatment of anal HSIL significantly reduces the risk of cancer progression [28], and the consequent recently published anal cancer screening recommendations for high-risk patients [29], there is a need to better understand the population of women at high-risk of anal cancer.

This study aims to investigate national anogenital HPV cancer trends as well as the role of demographics, deprivation, and ethnicity on anogenital cancer incidence in England, in an attempt to better understand this cohort of women which is increasingly affected by anogenital HPV-related disease.

## 2. Materials and Methods

### 2.1. Dataset

This is a cross-sectional study, which follows the ‘strengthening the reporting of observational studies in epidemiology’ (STROBE) statement [30]. National Cancer Registration and Analysis Service (NCRAS) data were accessed online via the CancerStats2 platform [31]. This is a secure platform which enables users to generate reports using NCRAS data on a self-service basis. Cancer Outcomes and Services Dataset (COSD) reports, which are submitted to NCRAS by all the NHS providers of cancer services in England on a monthly basis since January 2013 [32], are available on this platform and were accessed for the purpose of this study. The Level 3 COSD curated data function was utilised; this function allows for the creation of anonymised bespoke incidence and staging datasets using filters based on cancer registration records from 2014–2020, with data from 2013 no longer being accessible on the CancerStats2 platform [31].

Anonymised patient level data on all female patients above the age of 25 years with vulval, cervical, vaginal, and anal cancer between January 2014 and December 2020 were extracted; this included data on age at diagnosis, ethnicity, deprivation score and stage at diagnosis.

Age at diagnosis was classified into 5-year intervals. With respect to ethnicity, COSD utilises the 16 + 1 ethnic data categories defined in the 2001 census, which is the national mandatory standard [33]. Deprivation scores are based on the Index of Multiple Deprivation (IMD), this is the official measure of deprivation in England [34]. The index of deprivation (IoD) comprises seven distinct domains of deprivation, which when combined and appropriately weighted make up the IMD. These domains are: income, employment, health deprivation and disability, education skills training, crime, barriers to housing and services, and living environment, carrying weightings of 22.5%, 22.5%, 13.5%, 13.5%, 9.3%, 9.3%, 9.3%, respectively, in the IMD [34]. The IMD is expressed in quintiles, with quintile 1 being the most deprived and quintile 5 being the least deprived. COSD staging for anal cancer is based on the 8th edition of the Union for International Cancer Control TNM (tumour, nodes, metastasis) Classification of Malignant Tumours (UICC TNM8) [33,35]. Cervical, vaginal and vulval cancer staging is based on the International Federation of Gynaecology (FIGO) classification FIGO 2018, 2009 and 2021, respectively [33].

### 2.2. Incidence Rates

Incidence rates were calculated per 100,000 people in England per year using data extracted from the Office of National Statistics (ONS) website [36]. Population estimates for the general population broken down by age, sex, ethnicity and deprivation were downloaded and used to calculate crude incidence rates as well as age specific, ethnicity specific, deprivation specific and staging specific incidence rates. The crude rate equals the total number of new cancer cases diagnosed in a specific year in the population category of interest, divided by the at-risk population for that category, multiplied by 100,000 [37]. Category-specific rates are calculated by dividing the number of cancer cases diagnosed in a specific year in a specified category group by the total population of that category group for that year, multiplied by 100,000 [37]. Of note, ONS ethnicity data are not available for 2020; therefore, ethnicity specific incidences are not available for 2020. For incidence data, the median incidence with the upper lower quartiles are reported.

To investigate incidence trends between 2014 and 2020, we calculated the Average Annual Percentage Change (AAPC). This involved first calculating the annual percentage change in incidence between each consecutive year using the formula:((Incidence Later year − Incidence Earlier year)/Incidence Earlier year) × 100

The AAPC was then obtained by averaging these annual percentage changes. The mean and standard deviation were selected for this and are presented in the tables in Appendix A. Of note, it was not possible to carry out any statistical analysis on these trends. Multiple regression models, including linear and join-point regression, were trialled; however, due to the small number of data points, these methods were not feasible.

### 2.3. Statistical Analysis

Statistical analysis was conducted using R version 4.2.3. Plots were produced using R-package ggplot2 and frequency tables constructed using R-package table1. Frequency tables, bar charts, and line plots were produced to explore patterns and relationships between the data categories. Frequency tables were used to analyse the distribution of the categorical variables, whilst bar charts and line plots were used to visually demonstrate distributions, in turn, illustrating any trends over time. Chi-square or Fisher’s exact tests were carried out to assess relationships between the categorical variables. Chi-square results are reported as:X^2^ ((degrees of freedom, *n* = sample size) = chi-square value, *p* = *p*-value)

A statistically significant *p* value of <0.05 was set for this analysis. Multivariate logistic regression analysis was additionally carried out to further assess the relationships between different variables. For the purpose of this analysis, age was coded as below or above 55 years with age < 55 years being set as the reference group. Deprivation was coded as high (quintiles 1–2) and low (quintiles 4–5) with low deprivation being set as the reference group. For ethnicity, white ethnicity was set as the reference group, and staging was coded as early (stage 1–2) and late (stage 3–4) stage. Adjusted odds ratios (ORs) for the outcome of interest were obtained by accounting for the presence of specific characteristics while controlling for the effects of other factors. The odd’s ratio (OR), 95% confidence interval (CI) and *p*-value are reported.

For analysis, age groups were further grouped into: <40, 40–54, 55–74, 75+ years. The ‘missing values’ category and staging were excluded from all analysis.

## 3. Results

### 3.1. Basic Demographics

Between January 2014 and December 2020, 32,893 anogenital cancers were diagnosed in women over the age of 25 years. Of these, 17.8% (*n* = 5852) were anal cancer, 54.3% (*n* = 17,854) cervical cancer, 4.4% vaginal cancer (*n* = 1433) and 23.6% (*n* = 7754) vulval cancer diagnoses. A total of 32.2% (*n* = 10,606) of these women were aged between 55–74 years, 89.2% (*n* = 29,354) were of white ethnicity, 24.2% (*n* = 7952) were in the lowest deprivation 1 quintile and 46.2% (*n* = 10,242) of diagnoses were made at stage 1 disease. See Appendix A (Appendix A).

### 3.2. General Anogenital Cancer Incidence Trends

There has been a general growth in the total number of anogenital cancers between 2014 (*n* = 4537) and 2019 (*n* = 4972), with a decline being seen between 2019 and 2020; see Figure 1 and Appendix A (Appendix A).

The average annual percentage changes (AAPC) for anal, cervical, vaginal and vulval cancer incidences are 1.7%, −1.7%, −0.5% and −1.0%, respectively; see Figure 1 and Appendix A (Appendix A).

Cervical cancer is still the most prevalent malignancy, with a median incidence rate of 12.9 per 100,000 people. This is followed by vulval, anal and vaginal cancers, with median incidence rates of 5.6, 4.2, and 1 per 100,000 people, respectively.

### 3.3. Anogenital Cancer Trends and Age

The highest number of overall anogenital cancers is seen in the 55–74 year age group, peaking at 1495 diagnoses in 2019, see Figure 2.

Cervical cancer is most prevalent in the <40 years group (median incidence 18.3 per 100,000 people), anal cancer in the 55–74 and 75+ year groups (median incidences 7.3 and 8.0 per 100,000 people, respectively) and vulval and vaginal cancer in the 75+ group (median incidence 18.5 and 2.8 per 100,000 people, respectively); see Appendix A (Appendix A).

The incidence of cervical cancer has shown an average annual decrease of −3.4% and −5.4% in the <40 and 75+ years age groups, respectively, and an average annual increase of 2.5% and 0.6% in the 55–74 and 40–54 year age groups, respectively; see Figure 2 and Appendix A (Appendix A). The incidence of vulval cancer has shown an average annual increase of 18.2% in the <40 years age group and decrease of −1.0%, −0.2% and −1.9% in the 40–54, 55–74, 75+ year age categories. That of vaginal cancer has had an average annual increase of 25.4%, 2.3% and 2.4% in the <40, 40–54 and 55–74 year age groups, with the 75+ year group seeing an average annual decrease of −2.2%. Anal cancer incidence has been increasing in all age categories, with the largest AAPC in the <40 years group at 12.7%. See Figure 2 and Appendix A (Appendix A).

### 3.4. Anogenital Cancer Trends and Ethnicity

Anogenital HPV cancers in women have been rising most steeply in the Black and Mixed ethnicity groups, see Figure 3. Cervical cancer has the highest incidence in the Mixed and Unknown ethnicity groups, with median incidences of 23.2 and 84.1 per 100,000 people; see Appendix A (Appendix A). These appear to be decreasing (AAPC = −2.3%) in the Unknown ethnicity group but increasing (AAPC = 10.5%) in the Mixed ethnicity group, which carries the highest average annual increase for cervical cancer; see Figure 3 and Appendix A (Appendix A). Vaginal cancer has the highest incidence in the Unknown ethnicity group (median incidence 2.5 per 100,000 people); however, the largest average annual increases in incidence were seen in the Asian and Mixed ethnicity groups at 62.4% and 33.4%, respectively. The median incidence of vulval cancer is highest in the Unknown ethnicity group at 22.5 per 100,000 people. Asian ethnicity is the only group in which the average annual incidence of vulval cancer has shown an increase (AAPC = 18.1%). Anal cancer median incidence is highest in the Unknown ethnicity group at 15.5 per 100,000 people and has been increasing in all groups, with the largest AAPC being seen in Black and Mixed ethnicities at 118.8% and 17.0%, respectively. See Figure 3 and Appendix A (Appendix A).

### 3.5. Anogenital Cancer Trends: Age and Ethnicity

Cervical cancer is the most prevalent malignancy in the <40 years age group for all ethnicities, with the highest incidences seen in Mixed, Unknown, and White ethnicities with median rates of 17.1, 77.8 and 20.2 per 100,000 people, respectively; see Appendix A (Appendix A). Cervical cancer incidences for this age group have been on the decrease in Asian, Unknown and White ethnicities with AAPC of −0.9%, −3.1% and −7.1%, respectively. For Black and Mixed ethnicities, the AAPC demonstrate rises of 9.7% and 0.5%, respectively; see Appendix A (Appendix A) and Figure 4. Interestingly, whilst the incidence of cervical cancer is highest in the <40 y age group, as seen in Figure 2 and Appendix A (Appendix A), when the data are broken down by ethnicity, this is only true for White ethnicity; Asian, Black and Unknown ethnicities have the highest incidences in the 55–74 y and 75 y+ age groups; see Appendix A (Appendix A).

Cervical cancer remains the most prevalent in the 40–54 age group for all ethnicities, with the highest median incidences seen in Mixed, Unknown, and White ethnicities with rates of 27.9, 72.6 and 12.8 per 100,000 people, respectively; see Appendix A (Appendix A). In this age group, cervical cancer incidences have been on the rise for all ethnicities except White, with the largest AAPC of 34% being seen in Mixed ethnicity. For vulval cancer, the highest median incidences are seen in Unknown and Mixed ethnicities with rates of 10.6 and 4.1 per 100,000 people, respectively. These are decreasing in the Black ethnicity group (AAPC = −2.7%) and increasing in Asian, Mixed, Unknown and White groups (AAPC = 79.6%, 36.0%, 12.9%, and 2.2%, respectively). The highest median anal cancer incidence is in the Unknown group at 8.5 per 100,000, and so was the largest AAPC at 62.4%; see Appendix A (Appendix A).

In the 55–74 year age group, the gaps in incidences between cervical, vulval and anal cancer are smaller. Whilst cervical cancer is still the most prevalent cancer in all the ethnicities, in the White ethnicity group, vulval and anal cancer incidences have surpassed that of cervical cancer in 2019 (8.9 and 8.0 vs. 7.6 per 100,000 people, respectively), see Figure 4. In the Asian group, vulval cancer has had an average annual increase of 66.6% with its incidence in 2019 nearing that of cervical cancer (6.3 vs. 6.8 per 100,000 people). Cervical cancer incidence has seen an average annual percentage decrease of −1.7% and −3.7% in Asian and Unknown groups, respectively, and the largest average annual percentage rise was seen in the Mixed group at 22.4%. Cervical, vulval, vaginal and anal cancer median incidences are highest in the Unknown group at 88.1, 64.2, 8.6 and 52.8 per 100,000 people, respectively. Anal cancer incidence has been rising in all ethnicities, with the largest AAPC in the Black ethnicity group at 132.9%.

In the 75+ years age group, cervical cancer is the most prevalent pathology for Asian and Black ethnicities (median incidences of 11.8 and 20.6 per 100,000 people, respectively). The AAPC shows increases in the incidence of cervical cancer for all but White ethnicity groups, with the highest AAPC being seen in Black and Mixed groups at 230.7% and 73.5%, respectively. In Mixed, Unknown and White ethnicities, median vulval cancer incidence is the highest at 53.1, 310.2 and 18.1 per 100,000 people, respectively. The highest AAPC is seen in Asian ethnicity at 18.2%. Anal cancer incidence has shown an average annual decrease of −4.5% in White ethnicity and the biggest AAPC was seen in Mixed ethnicity at 51.6%. The highest median anal cancer incidence is seen in the Unknown group at 121.4 per 100,000 people.

### 3.6. Anogenital Cancer Trends and Deprivation

The highest median incidences for all cancers are in the deprivation 1 (most deprived) quintile with incidences of 4.9, 18.3, 1.3 and 6.3 per 100,000 people for anal, cervical, vaginal and vulval cancer, respectively. Conversely, the lowest median incidences for all cancers are in the deprivation 5 (least deprived) quintile with median incidences of 3.8, 9.7, 0.9 and 4.9 per 100,000 people for anal, cervical vaginal and vulval cancer, respectively. The incidences of these cancers are therefore higher in deprived vs. not deprived populations, see Appendix A (Appendix A). Cervical and vulval cancer incidences have seen decreasing AAPC in all quintiles, whilst those for vaginal and anal cancer have seen increasing AAPC in all quintiles. The AAPC in anal cancer incidence are highest in the in the 1st and 5th quintiles at 2.9% and 6.7%. See Figure 5 and Appendix A (Appendix A).

### 3.7. Anogenital Cancer Trends: Age and Deprivation

When looking at deprivation and age specific incidence trends, similar patterns to those seen in the deprivation-specific and age-specific trends are noted. The median incidences of all anogenital cancers are highest in the deprivation 1 quintile and lowest for the deprivation 5 quintile for all age groups; see Appendix A (Appendix A) and Figure 6. In addition to this, cervical cancer has the highest incidences in all deprivation quintiles in <40 y, 40–50 y and 55–74 y age groups, whilst vulval, anal and cervical cancers have the highest incidences in the 75+ y, the 55–74 y and 75+ y and 75+ y group for all deprivation groups, respectively; see Appendix A (Appendix A) and Figure 6.

More specifically, the highest median incidence for anal cancer is seen in the deprivation 1 quintile and 75+ y age group at 10.2 per 100,000 people, for cervical cancer in the deprivation 1 quintile and 40–54 y age group at 20.9 per 100,000 people, for vaginal cancer in the deprivation 1 quintile and 75+ y age group at 4.5 per 100,000 people and for vulval cancer in the deprivation 2 quintile and 75+ y age group at 20.3 per 100,000 people; see Appendix A (Appendix A).

### 3.8. Anogenital Cancer Trends and Staging

The majority (*n* = 10,242, 46.2%) of anogenital cancer diagnoses are made at stage 1 disease, with 66.1% (*n* = 14,640) of cases being diagnosed at stage 1 and 2 diseases and 33.8% (*n* = 7506) at stage 3 and 4 diseases. This number is driven by the number of cervical and vulval cancer diagnosis made at stage 1 (*n* = 5592, 51.9% and *n* = 3753, 65.2% of cervical and vulval cancer diagnoses, respectively), see Appendix A (Appendix A).

There has been a rise in stage 3 (AAPC = 34.4%) and stage 1 (AAPC 7.7%) disease for cervical cancer between 2014 and 2020, with a steep increase between 2019 and 2020 for both stages; see Figure 7 and Appendix A (Appendix A).

For vulval cancer incidence, there was an average annual increase of 2.5% and 5.3% for stage 3 and 4 diagnoses; see Figure 7 and Appendix A (Appendix A).

Vaginal cancer diagnoses are evenly split between different stages (23.9%, 21.3%, 23.4% and 31.4% at stage 1, 2, 3 and 4, respectively), with the highest incidence rise (AAPC 5.7%) being seen for stage 3 disease; see Figure 7 and Appendix A (Appendix A).

Most anal cancer diagnoses are made at stage 3 disease (*n* = 2161, 46.2%), with an average annual percentage increase of 9.3% in stage 3 diagnoses and a decrease of −1% in stage 1 diagnoses. See Figure 7 and Appendix A (Appendix A).

### 3.9. Anogenital Cancer Trends: Staging and Age

For anal cancer, the incidence of stage 3 disease is highest and increasing among all age groups, with the highest AAPC of 99.7% in the <40 years age group, see Appendix A (Appendix A) and Figure 8. The incidence of stage 1 disease shows a decreasing AAPC of −3.4% in the 55–74 y group, the AAPC have otherwise been increasing in the other age groups. The incidence of stage 4 disease is on the rise in all age groups, with the highest incidence in the 75+ years age group (median incidence 0.8 per 100,000 people) and highest AAPC (7.0%) in the 40–54 y age group.

For cervical cancer, the incidence of stage 1 disease is highest in the <40 years and 40–54 years age groups (median incidence 7.2 and 4.9 per 100,000 people, respectively); see Appendix A (Appendix A) and Figure 9. In the 55–74 y and 75+ age groups, stage 2 and 4 diagnoses have the highest median incidences. Stage 3 disease has some of the lowest incidence rates, but also the biggest increases in AAPC in all age categories.

For vulval cancer, stage 1 disease incidence is highest in all age categories, followed by stage 3 incidence, see Appendix A (Appendix A) and Figure 10. The incidence of stage 1 disease is on the rise in the <40 years and 40–54 years age groups and declining in the 55–74 and 75+ years age groups, whilst that of stage 3 has increased in all but the 75+ y group.

Vaginal cancer incidences, especially in the <40 y and 40–54 y age groups, are very low and difficult to reliably interpret. Generally, stage 4 incidence is highest in the 55–74 y and 75+ y age groups and the largest annual average percentage increases are seen in stage 3 disease for all age groups except the 75 y+ group; see Appendix A (Appendix A).

### 3.10. Chi-Square, Fisher’s Exact and Multivariate Logistic Regression Analysis

#### 3.10.1. Age and Ethnicity

There is a significant relationship between ethnicity and age for all anogenital cancers (Fisher’s exact test, *p* < 0.001); see Appendix A (Appendix A).

For cervical cancer Asian and Black ethnicities are associated with age groups over 40 years, whilst Mixed, Unknown and White ethnicities below 55 years of age, with White ethnicity also having an association with the 75+ age group.

For anal cancer, Asian and Unknown ethnicities are associated with age groups over 55 years, whilst White ethnicity is associated with over the age of 75 years. Mixed and Black ethnicities are associated with an age of under 55 years.

For vaginal cancer, Asian and Black ethnicities are associated with age groups under 55 years and Unknown and White ethnicities with age groups over 55 years.

For vulval cancer, Asian and Mixed ethnicities are associated with age groups 40–54 and 55–74 years, Black ethnicity with age groups under 55 years, Unknown with age < 40 and 55–74 years and White with age < 40 years and over 75 years.

Similar results were seen on logistic regression (see Appendix A (Appendix A)), where deprivation was controlled for.

For cervical cancer, patients of Asian and Black ethnicity are shown to be 33% (OR 1.33, CI 1.07–1.65, *p* = 0.009) and 58% (OR 1.58, CI 1.22–2.05, *p* ≤ 0.001) more likely to be over the age of 55 years compared to White patients, respectively. Patients of mixed ethnicity are shown to be 28% (OR 0.72, CI 0.75–0.91, *p* = 0.06) less likely to be over the age of 55 years compared to White patients, and there was no significant relationship with Unknown ethnicity.

For anal cancer, patients of Black and Mixed ethnicity are 70% (OR 0.30, CI 0.18–0.50, *p* < 0.001) and 55% (OR 0.45, CI 0.27–0.77, *p* = 0.003) less likely to be over the age of 55 years compared to White patients, respectively, whilst patients of Unknown ethnicity are 61% (OR 1.61, CI 1.06–2.54, *p* = 0.031) more likely to be over the age of 55 years compared to White patients, and there was no significant relationship with Asian ethnicity.

For vaginal cancer, patients of Asian and Black ethnicities were 75% (OR 0.25, CI 0.13–0.47, *p* < 0.001) and 65% (OR 0.35, CI 0.18–0.69, *p* = 0.002) less likely to be over the age of 55 years compared to White patients, respectively. No other significant relationships were noted.

For vulval cancer, patients of Black ethnicity are 80% (OR 0.20, CI 0.12–0.32, *p* < 0.001) less likely to be under the age of 55 years compared to White patients. No other significant relationships were noted.

#### 3.10.2. Staging and Ethnicity

There is a significant relationship between staging and ethnicity for cervical (X^2^ (12, *n* = 17,854) = 35.5, *p* < 0.001) and vulval (Fisher’s exact test, *p* < 0.05) cancers only, see Appendix A (Appendix A).

For cervical cancer, Asian ethnicity is associated with stage 4, Black ethnicity with stage 2–3, Mixed ethnicity with stage 3, Unknown with stage 1 or 4 and White with stage 2 disease.

For vulval cancer Asian ethnicity is associated with stage 2 and 3, Black with 1 and 4, Mixed with stage 2–4, Unknown with stage 2 and 4 and White with stage 1 and 3 diseases.

On logistic regression (see Appendix A (Appendix A)), with age and deprivation controlled for, significant relationships were seen for vaginal and vulval cancers only. For vaginal cancer, patients of Black ethnicity are 66% (OR 0.34, CI 0.15–0.75, *p* = 0.009) less likely to have a late-stage diagnosis than White patients.

For vulval cancer, patients on Unknown ethnicity are 56% (OR 1.56, CI 1.09, 2.21, *p* = 0.014) more likely to have a late-stage diagnosis than White patients.

#### 3.10.3. Deprivation and Ethnicity

There is a significant relationship between deprivation and ethnicity for all anogenital cancers. Anal cancer (X^2^ (16, *n* = 5852) = 47.5, *p* < 0.001), cervical cancer (X^2^ (16, *n* = 17,854) = 170.6, *p* < 0.001), vaginal cancer (Fisher’s exact *p* < 0.001), vulval cancer (X^2^ (16, *n* = 7754) = 80.9, *p* < 0.001), see Appendix A (Appendix A).

Asian, Black and Mixed ethnicities are associated with deprivation quintiles 1 and 2 for all anogenital cancers, aside from Mixed ethnicity for vaginal cancer, where there is no significant relationship. White and Unknown ethnicities are associated with deprivation quintiles 3 and 4.

On logistic regression (see Appendix A (Appendix A)), with age controlled for, similar relationships were found.

For cervical cancer, patients of Asian, Black, and Mixed ethnicities are 58% (OR 1.58, CI 1.26, 1.98, *p* < 0.001), 407% (OR 5.07, CI 3.51–7.59, *p* < 0.001) and 68% (OR 1.68, CI 1.36–2.10, *p* < 0.001) more likely to be highly deprived than patients of White ethnicity, respectively. Patients of Unknown ethnicity are 20% (OR 0.80, CI 0.69–0.92, *p* = 0.001) less likely to be highly deprived than White patients.

For anal cancer patients of Black ethnicity are 287% (OR 3.87, CI 2.09–7.85, *p* < 0.001) more likely to be highly deprived than White patients.

For vaginal cancer, patients of Asian and Black ethnicities are 437% (OR 5.37, CI 2.40, 14.4, *p* < 0.001) and 269% (OR 3.69, CI 1.76–8.70), *p* = 0.001) more likely to be highly deprived than White patients, respectively.

For vulval cancer, patients of Asian, Black and Mixed ethnicities are 99% (OR 1.99, CI 1.35–2.98, *p* < 0.001), 378% (OR 4.78, CI 2.55, 9.98, *p* < 0.001) and 147% (OR 2.47, CI 1.50, 4.25, *p* < 0.001) more likely to be highly deprived than White patients, respectively.

#### 3.10.4. Age and Deprivation

There is a significant relationship between deprivation and age for anal (X^2^ (12, *n* = 5852) = 53.8, *p* < 0.001), cervical (X^2^ (12, *n* = 17,854) = 104.1, *p* < 0.001) and vulval cancer (X^2^ (12, *n* = 7754) = 147.3, *p* < 0.001); see Appendix A (Appendix A).

For anal cancer, the age group <40 years is associated with deprivation quintiles 1–3, age 40–54 years with quintiles 1 and 2, age 55–74 years with quintiles 3–5 and age 75+ years with quintiles 4 and 5.

For cervical cancer, the age group <40 years is associated with deprivation quintiles 1–2, age 40–54 years with quintiles 1, age 55–74 years with quintiles 3 and 4 and age 75+ years with quintiles 3–5.

For vulval cancer, age group <40 years is associated with deprivation quintiles 1, age 40–54 years with quintiles 1 and 2, age 55–74 years with quintiles 4 and 5 and age 75+ years with quintiles 3–5.

On logistic regression (see Appendix A (Appendix A)), with ethnicity controlled for, similar relationships are noted.

Anal, cervical, and vulva cancer patients over 55 years of age are 30% (OR 0.70, CI 0.61–0.80, *p* < 0.001), 19% (OR 0.81, CI 0.75–0.87, *p* < 0.001) and 44% (OR 0.56, CI 0.49–0.64, *p* < 0.001) less likely to be highly deprived than patients under 55 years, respectively.

#### 3.10.5. Staging and Deprivation

There is a significant relationship between deprivation and staging for cervical (X^2^ (12, *n* = 17,854) = 53.2, *p* < 0.001) and vaginal (X^2^ (12, *n* = 1433) = 23.5, *p* < 0.05) cancers; see Appendix A (Appendix A).

For cervical cancer, stage 1 disease is associated with deprivation quintiles 3–5, stage 2 disease with quintiles 2 and 5, stage 3 disease with quintiles 1 and 2 and stage 4 disease with quintiles 1 and 3.

For vaginal cancer, stage 1 disease is associated with deprivation quintiles 4 and 5, stage 2 disease with quintiles 1, 2 and 5, stage 3 disease with quintiles 2 and 3 and stage 4 disease with quintiles 1 and 3.

On logistic regression, with age and ethnicity controlled for, similar relationships were seen (see Appendix A (Appendix A)). Patients with high deprivation are 44% (OR 1.44, CI 1.29–1.59, *p* < 0.001) and 38% (OR 1.38, CI 1.02–1.86, *p* < 0.001) to be diagnosed with late-stage cervical and vaginal cancer, respectively.

#### 3.10.6. Staging and Age

There is a significant relationship between age and staging, for anal cancer (X^2^ (9, *n* = 5852) = 55.4, *p* < 0.001), cervical cancer (X^2^ (9, *n* = 17,854) = 2025.4, *p* < 0.001) and vulval cancer (X^2^ (9, *n* = 7754) = 82.2, *p* < 0.001); see Appendix A (Appendix A).

For anal cancer, stage 1 disease is associated with age groups <40 and 55–74 years, stage 2 with group 75+ years, stage 3 with ages under 75 years and stage 4 with age group 75+ years.

For cervical cancer, stage 1 disease is associated with age groups under 55 years, stage 2 with age groups over 55, stage 3 with age groups over 40 years and stage 4 with ages above 55 years.

For vulval cancer, stage 1 disease is associated with age groups under 55 years, stage 2 and 3 with age groups over 55 years, and stage 4 with ages above 75 years.

On logistic regression, with ethnicity and deprivation controlled for, similar relationships were seen (see Appendix A (Appendix A)).

For anal cancer, women aged ≥ 55 years are 18% (OR 0.82, CI 0.70–0.96, *p* = 0.009) less likely to have a late-stage diagnosis than women under 55 years of age.

For cervical cancer, women aged ≥ 55 years are 233% (OR 3.33, CI 3.01–3.69, *p* < 0.001) more likely to have late-stage diagnosis than women under 55 years of age.

For vulval cancer, women aged ≥ 55 years are 80% (OR 1.80, CI 1.50–2.18, *p* < 0.001) more likely to have late-stage diagnosis than women under 55 years of age.

## 4. Discussion

The sexual revolution of the 1960s and 1970s promoted risky sexual behaviour, leading to an earlier age at first intercourse and promiscuity [14,38,39]. This had a significant impact on the prevalence of all sexually transmissible infections, including hrHPV and, consequently, its associated anogenital cancers [14,38,39]. This is reflected in our data, which demonstrate a general rise in the total number of new anogenital cancer diagnoses in England between 2014 and 2020, especially for women aged 55–74 years (see Figure 1 and Figure 2 and Appendix A (Appendix A)). Interestingly, this rise was met by a decline in total cancer numbers between 2019 and 2020, which is likely a consequence of missed cancer diagnoses secondary to the COVID-19 pandemic [40,41], rather than a true decline in new cancer diagnoses. The negative impact of the COVID-19 pandemic on cancer-related morbidity and mortality is widely reported [42,43] and beyond the aim of this article; however, it is important to not overlook its potential impact on this dataset, especially on the 2020 incidences which could consequently be outliers. The true trajectory of these incidence trends will be clarified with the release of post-pandemic data in the near future.

Despite this, our findings are still in keeping with those published in other international series [10]. More specifically, it is clear that the implementation of national cervical cancer prevention strategy has positively impacted cervical cancer incidence, which has been declining in the <40-year group, the age group most likely to be benefitting from both the introduction of HPV vaccination in 2008 and the improved performance of primary hrHPV testing (vs. cytology alone), which was implemented as part of cervical screening programme in England in 2016; see Appendix A (Appendix A). Although, it is worth noting that vaccinated patients would have only been 25 years of age in 2020, making it too early to truly assess the impact of vaccination on cervical cancer incidence. Nevertheless, as many as 51.9% of cervical cancer diagnoses were made at stage 1 disease, with the incidence of stage 1 being highest in women under the age of 55 years, an association which was significant on chi-square analysis (X^2^ (9, *n* = 17,854) = 2024.4, *p* < 0.001) and logistic regression (*p* < 0.001); see Appendix A (Appendix A).

The incidence of cervical cancer has, however, seen an average annual increase of 2.5% in the 55–74-year age group, with stage 3 and 4 diseases being significantly associated with age groups above 40 and 55 years, respectively (X^2^ (9, *n* = 17,854) = 2024.4, *p* < 0.001), an association which was also seen on logistic regression, which shows women over 55 years to be 233% (*p* < 0.001) more likely to have a late-stage diagnosis than women under 55 years; see Appendix A (Appendix A). Cervical cancer incidence in women over the age of 50 years is an important topic of discussion, with many countries noting a second peak of cervical cancer in women over the age 50 years [44]; theories underlying this phenomena revolve around re-exposure to hrHPV from new sexual partners in mid-life, and the re-activation of a latent HPV infection with age-related immunosuppression [44]. It is also important to note that these women would have been screened and treated for cervical dysplasia since 1964, in turn delaying the age at which they would develop cervical cancer. Consistent with this, Kalliala et al. [16] found women treated for cervical dysplasia to have a cervical cancer incidence of 39 per 100,000 people (vs. 12.9 per 100,000 people in the general population, as per this dataset) and a more elevated relative risk of cervical cancer compared with the general population (3.34, 2.39–4.67, *p* < 0.001), a risk which was higher in women over the age of 50 years and which remained elevated for at least 20 years after treatment. This raises discussions over whether the age of cervical screening should be extended past 65 years, especially with studies demonstrating that, over the next 25 years, the cervical cancer burden will increasingly be affecting women over the age of 50 years [45].

On a similar note, the rates of vulval and anal cancer which typically occur in women over the age of 50 years have also been reported to be on the rise [9,25]. A study looking at vulval cancer incidence in high-income countries found a 4.6% general increase in vulval cancer incidence between 1988–1992 and 2003–2007 [9]; our study found an average annual percentage decrease of −1.0%; see Figure 1 and Appendix A (Appendix A). However, it is important to note that our dataset only spans 7 years. Of note, we found an 18.2% average annual percentage rise in the incidence of vulval cancer in women <40 years of age. This is in agreement with other studies which also found vulval cancer rates to increase in women under the age of 60 years [9]. The incidence of vulval cancer in women under 60 years is thought to reflect the incidence of HPV positive vulval cancers, as opposed to HPV negative cancers arising from lichen sclerosis-related differentiated vulval intraepithelial neoplasia (dVIN) [46,47,48]. This is because HPV-related usual vulval intraepithelial neoplasia (uVIN) is also on the rise [9] and is typically found in younger women [46,47,48], with women < 55 years of age being three times more likely to have HPV DNA positive vulval disease than women over the age of 65 years [49].

Anal cancer incidence trends in women are arguably the most remarkable. This study found a 1.7% average annual percentage increase in anal cancer incidence between 2014–2020; see Appendix A (Appendix A) and Figure 2. The rising burden of anal cancer in women has been documented in numerous publications [10,25,26,50,51]. Two thirds of all anal cancer diagnoses are in women [25], and unfortunately women are presenting with advanced anal disease. In this study, 43.2% of women in England presented with stage 3 disease, with only 14.4% presenting with stage 1 disease; see Appendix A (Appendix A). Moreover, the incidence of stage 3 disease saw the biggest average annual rise at 9.3%, whilst the incidence of stage 1 disease saw a fall of −1.0%; see Appendix A (Appendix A) and Figure 2. This is clearly an issue, given that survival rates are significantly better for stage 1 and 2 compared to stage 3 and 4 diseases: 90% and 80% 5-year survival rates versus 60% 5-year and 55% 1-year survival rates, respectively [52]. These findings welcome the current drive towards the introduction of anal cancer screening in high-risk populations [29], especially high-risk women. More clearly needs to be done to improve the earlier presentation and detection of anal cancer in women. This is especially true for White women aged 55–74 years, where the incidence of anal cancer surpassed that of cervical cancer (8.0 vs. 7.6 per 100,000 people, respectively); see Figure 4. This pattern was also identified by Deshmukh et al. [10], who analysed American Surveillance, Epidemiology and End Results (SEER) data; they found the incidence of anal cancer for white women aged 65–74 years exceeded that of cervical cancer in 2015, with incidences of 8.6 and 8.2 per 100,000 people, respectively.

One of the aims of this study was to explore the role of certain sociodemographic risk factors on the trends of these cancers. We found significant associations between age at diagnosis and staging, with younger age groups (<55 years) being associated with earlier staging for cervical, anal and vulval cancer, on chi-square analysis (see Appendix A (Appendix A)). For cervical cancer, this is likely associated with screening, for vulval and anal cancer the cause of this relationship is less clear. Interestingly, for anal cancer, on logistic regression, once ethnicity and deprivation were controlled for, women under the age of 55 years were more likely to be associated with late-stage disease than women over the age of 55 years (see Appendix A (Appendix A)). Studies exploring the relationship between age and delayed cancer diagnoses found significant relationships between longer diagnostic interval and increasing age [53,54]; others found elderly patients more likely to present with advanced disease [55]. Age is therefore an important risk factor for delayed presentation with advanced disease. This could in turn be a reflection on our health services and how accessible they truly are for our more elderly patient populations, as well as a lack of education around the risk of all anogenital HPV-related pathology later on in life [56]. Other possibilities may revolve around younger women being more likely to self-examine or more likely to be under surveillance for other HPV-related pathologies, resulting in earlier lesion identification. This does not, however, explain the relationship between age < 55 years and late-stage anal cancer diagnosis, which could perhaps be related to diagnostic uncertainty and misdiagnosis; anal cancer is often asymptomatic, difficult to diagnose and confused for other anorectal pathology such as haemorrhoids, especially in younger patients [27,57,58].

Ethnicity is another important factor. In our data, 89.2% of all anogenital cancer patients were of White ethnicity—see Appendix A (Appendix A)—this is consistent with the fact that 81.7% of the population in England and Wales is in fact White [36]. The total numbers have, however, increased most steeply in the Black and Mixed ethnicity groups; see Figure 3. Moreover, incidences for all anogenital cancers were highest in the Unknown ethnicity (84.1, 2.5, 22.5 and 15.5 per 100,000 people for cervical, vaginal, vulval and anal cancer, respectively), with cervical cancer incidence increasing namely in the Mixed ethnicity group (AAPC = 10.5%), vaginal cancer incidence showing the highest average annual rises in Asian and Unknown ethnicity groups (AAPC = 62.4% and 33.4%, respectively), vulval cancer incidence increasing solely in the Asian group (AAPC = 18.1%) and anal cancer incidence showing the highest rises in Black and Mixed groups (AAPC = 118.8% and 17.0%, respectively); see Appendix A (Appendix A). It is important to note that Unknown ethnicity in COSD includes: ‘Chinese’, ‘any other ethnic group’, ‘not stated’ and ‘not known’ [33]. Whilst we are unable to separate the proportions of these ethnicities given the nature of the dataset, research looking at the concordance between administrative data on race and ethnicity and patient self-reported race and ethnicity show poorer concordance for ethnic groups other than ‘white’, with ethnic minority groups showing agreement with categories such as ‘other’ on administrative data [59,60,61]. There is therefore clearly a considerably high HPV-related disease burden in ethnic minority groups, which cannot be ignored.

When exploring the relationship between ethnicity and age, there was a significant relationship for all anogenital cancers, White patients with anal, vaginal and vulval cancers were significantly more likely to be over the age of 55 years compared to other ethnicities. The opposite was true for White patients with cervical cancer, who were significantly more likely to be under the age of 55 years compared to other ethnicities, as seen on logistic regression; see Appendix A (Appendix A). This reflects our results, also demonstrating that the incidence of cervical cancer in non-white ethnicities is highest in age groups over 55 years; see Appendix A (Appendix A). Furthermore, for cervical and vulval cancer, chi-square analysis revealed White ethnicity to be the only group significantly associated with stage 1 disease, with stage 4 disease being significantly associated with Asian and Unknown groups for cervical and Black, Mixed and Unknown for vulval cancer. Interestingly, when age and deprivation were controlled for, a similar relationship was true for vulval and vaginal cancer only, where patients of Black and Unknown ethnicities are shown to be more likely to present with late-stage disease than White patients, respectively (see Appendix A (Appendix A)). This suggests that for cervical cancer, deprivation may be a more important factor for late-stage presentation than ethnicity. Women of ethnic minority groups are therefore presenting at relatively younger (for anal, vaginal and vulval cancer) or older (for cervical cancer) ages with more advanced disease compared to women of White ethnicity, at least for vulval and vaginal cancer. Similar associations have been reported elsewhere for cervical and anal cancer, with Black women having higher incidences of disease [25,62,63], higher cancer mortality [63,64,65] and lower compliance with treatment [25,66].

The reasons for this are likely extremely complex; however, they are possibly linked to the social stigma and embarrassment attached to HPV disease, which can delay presentation when symptoms first arise. This has been shown to be particularly true for certain ethnic groups; with studies highlighting a general fear of death associated with a positive HPV test, as well as fear of discrimination, secondary to assumptions of promiscuity and infidelity [67,68,69,70,71]. Ethnic minorities have been shown have a lower awareness of their cancer risk as well as a poor understanding of HPV disease and the purpose of screening programmes [56,67,69], which in turn translates to a delayed diagnosis and poorer prognosis. A study in Finland found that among migrant women of African origin, language difficulties and a lack of screening information were the main barriers to cervical screening participation [72].

Intricately linked to this is socioeconomic status, with high deprivation being associated with difficulties in accessing healthcare due to lower education levels and cancer risk awareness [73], as well as unstable employment and housing affecting one’s ability to attend appointments and receive hospital correspondence [74]. We found the burden of HPV-related anogenital cancers to be highest in deprivation quintile 1, the most deprived quintile; with the highest median incidences seen in this quintile for all anogenital cancers (4.9, 18.3, 1.3 and 6.3 per 100,000 people for anal, cervical, vaginal and vulval cancer, respectively); see Appendix A (Appendix A). More importantly, there was a significant relationship between ethnicity and deprivation for all cancers, with Asian, Black, Unknown and Mixed ethnicities being more likely to be associated with high deprivation (deprivation quintiles 1 and 2) than White ethnicity, as seen on logistic regression; see Appendix A (Appendix A). In addition to this, for cervical and vaginal cancer, high deprivation (quintiles 1 and 2) was significantly associated with advanced staging (stage 3 and 4), whilst low deprivation (quintiles 3–5) was associated with early staging (stage 1 and 2); see Appendix A (Appendix A). This relationship was also seen on logistic regression where ethnicity and age were controlled for; see Appendix A (Appendix A). This is in concordance with other research, which also describes the negative impact of high deprivation on the incidence and prognosis of HPV related cancers, as well its relationship with certain ethnic minority groups [25,50,62,75,76,77]. There are undoubtedly complex interactions between ethnic minority groups, their cultural beliefs and the consequent impact of their deprivation (from a financial and educational perspective) which all together act as a barrier to healthcare, leading to delayed presentation, a more advanced disease at diagnosis and consequently, poorer prognosis. Linked to this is the fact that we also found significant relationships between deprivation and age for all but vaginal, cancers, with patients under 55 years of age being significantly (*p* < 0.001) more likely to be deprived than patients over the age of 55 years, as seen on logistic regression analysis; see Appendix A (Appendix A). The reasons for this are uncertain; however, they are likely due to significant cultural, economic or educational barriers to screening and presentation when new symptoms arise [55,78]; if patients do not comply with cancer prevention strategy, then they will be more likely to present at a younger age with a more advanced disease.

### Strengths and Limitations

This study reports a large dataset of anogenital cancers in England from 2014–2020 with a focus on demographics, ethnicity and social deprivation. There are, however, limitations to the COSD dataset which need to be acknowledged.

COSD reports are submitted to NCRAS by all NHS providers of cancer services in England on a monthly basis. There is no way of reviewing or authenticating the dataset for, for example, recall bias or duplicate entries. Moreover, it lacks data on end outcomes or on other important risk factors relevant to HPV-related disease, including HIV status, smoking, cervical screening results and vaccination status. The only way to link such data is to submit a data access release application (DARS) via NHS Digital, which carries a fee, requires ethics approval and is not freely available. We hope that the mASCARA database which was launched in 2019 will help overcome some of the issues around data access [79].

COSD uses ICD-10 coding, which specifies the site of the cancer but does not distinguish between HPV positive and negative disease nor the morphology of the cancer. This is not so much an issue for anal, cervical and vaginal cancer, of which 90%, 100% and 70% is caused by hrHPV. It does, however, present an issue for vulval cancer, of which only 40% is caused by hrHPV. Up to 60% of the patients included in the vulval cancer cohort are therefore likely to have HPV negative disease, which is not representative of the patient cohort we are aiming to study. Nevertheless, for the reasons mentioned in the discussion, the trends of vulval disease in women under 60 years of age could be used as a surrogate for HPV positive vulval disease.

Another limitation is the fact that the definition of ‘anal cancer’ changed in 2018 with the AJCC 8th Edition. Before 2018, perianal cancers were classified under the ICD-10 code 44.5, which coded for skin cancers of the trunk. These cancers will therefore not have been included under the ICD-10 21.0 anal cancer code and would have been excluded from the database before 2018.

With respect to ONS, ethnicity data for the year 2020 has not been released yet; therefore, ethnicity-specific incidence rates are limited to 2014–2019, unlike the rest of the data which is for 2014–2020. ONS also uses ethnic data categories as defined by the 2021 census [80], which differ slightly from those used by COSD, of which ethnic data categories are defined by the 2001 census [33]. Our categories comply with those in the 2001 census as per COSD.

In view of the limited time points of our dataset (seven time points between 2014 and 2020), it was not possible to carry out incidence trend testing. Multiple regression models including linear and join-point regression were trailed; however, due to the small number of data points, these methods were not feasible. In view of this, the AAPC, as described in this paper, was deemed the most appropriate method of presenting and describing the trend data. This paper therefore describes average annual percentage changes in the incidences of these anogenital cancers between 2014 and 2020. We acknowledge that data outliers have a bigger effect on datasets with limited timepoints, therefore our incidence trends need to be interpreted with caution. We, however, believe that these trends are still important and informative despite these limitations.

## 5. Conclusions

This study demonstrates that anogenital cancers are on the rise and carry a significant disease burden, especially for women over the age of 55 years. Our results show that this patient cohort is significantly more likely to present with late-stage cervical and vulval cancer compared to women under the age of 55 years. Moreover, in white women aged 55–74 years, the incidence of anal and vulval cancer has now overtaken that of cervical cancer.

Ethnicities other than White appear to be at a particularly high risk of these anogenital cancers, with the highest incidences being seen in the Unknown ethnicity group for all cancers, with cervical cancer incidence increasing namely in the Mixed ethnicity group, vaginal cancer incidence showing the highest average annual rises in Asian and Unknown groups, vulval cancer incidence increasing solely in the Asian group and anal cancer incidence showing the highest rises in Black and Mixed groups. Moreover, patients of Black and Unknown ethnicities with vaginal and vulval cancer are shown to be more likely to present with late-stage disease than White patients, respectively.

Intertwined with this is socioeconomic deprivation, which in England is a significant population risk factor for HPV-related disease. In fact, we found the burden of these cancers to be highest in deprivation quintile 1, the most deprived quintile, with high deprivation being significantly associated with younger age at diagnosis, being of non-white ethnicity and presenting with late-stage disease.

More clearly needs to be done to protect women over the age of 55 years from all anogenital cancers. These women are not benefitting from the protective effects of vaccination, with no woman born before 1991 benefitting from this in the next 25 years [45]. Furthermore, they are being screened and treated for cervical dysplasia, but are then at higher risk of developing cervical, vulval and anal cancer later on life. Cervical screening ends at the age of 65 years, and women are then not monitored or even educated about their risk of developing further HPV-related anogenital dysplasia. The introduction of anal cancer screening in high-risk women over 45 years of age, or 1 year after vulval HSIL diagnosis [29], could potentially help bridge the current gap created by the termination of the cervical screening programme at 65 years; however, this is likely only part of the solution, especially as the success of such a programme revolves around the availability of the centres able to provide high-resolution anoscopy, which are currently scarce. Moreover, this does not tackle the issues around education and access to healthcare, which are important barriers linked to the observed increased risk of anogenital cancers in patients with high-deprivation and of ethnicity other than White; attitudes towards screening and vaccination have all been shown to be affected by ethnicity and economic status [66,78,81,82]. If we are to try to reduce anogenital cancer incidence rates and improve rates of early disease detection, there is a need to tackle these attitudes, normalise HPV-related disease, and improve access to healthcare for those most deprived. Moreover, there is an argument for starting to manage anogenital HPV-related disease in women as one entity which affects women at different stages of life, rather than separate conditions defined by anatomy.

## Figures and Tables

**Figure 1 cancers-16-02177-f001:**
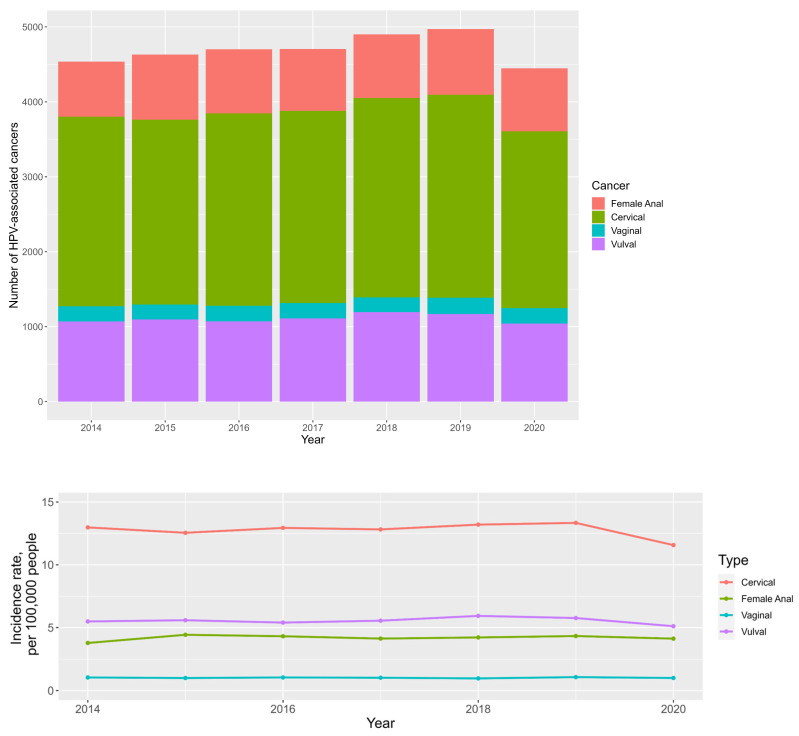
Stacked bar chart looking at total numbers of anogenital cancers between 2014 and 2020 (**top**). Line graph demonstrating anogenital cancer incidence trends in women over the age of 25 years between 2014 and 2020 (**bottom**).

**Figure 2 cancers-16-02177-f002:**
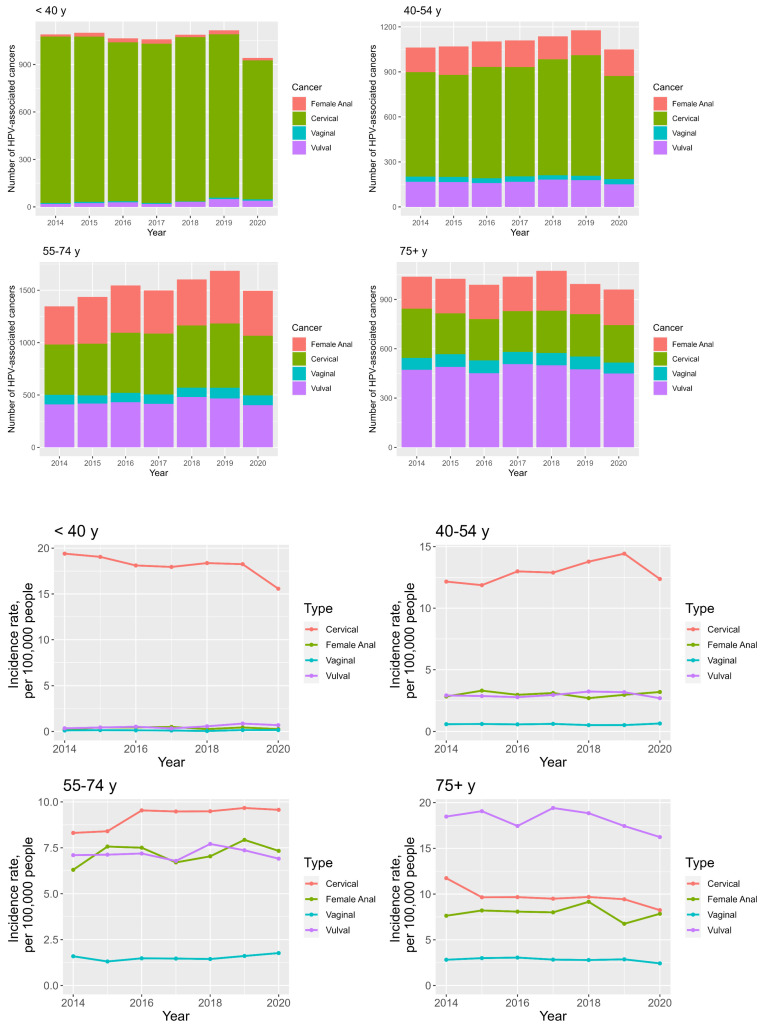
Stacked bar chart looking at the total numbers of anogenital cancers in women between 2014 and 2020, broken down in age groups (**top**). Line graphs demonstrating age—specific anogenital cancer incidence trends in women between 2014 and 2020 (**bottom**).

**Figure 3 cancers-16-02177-f003:**
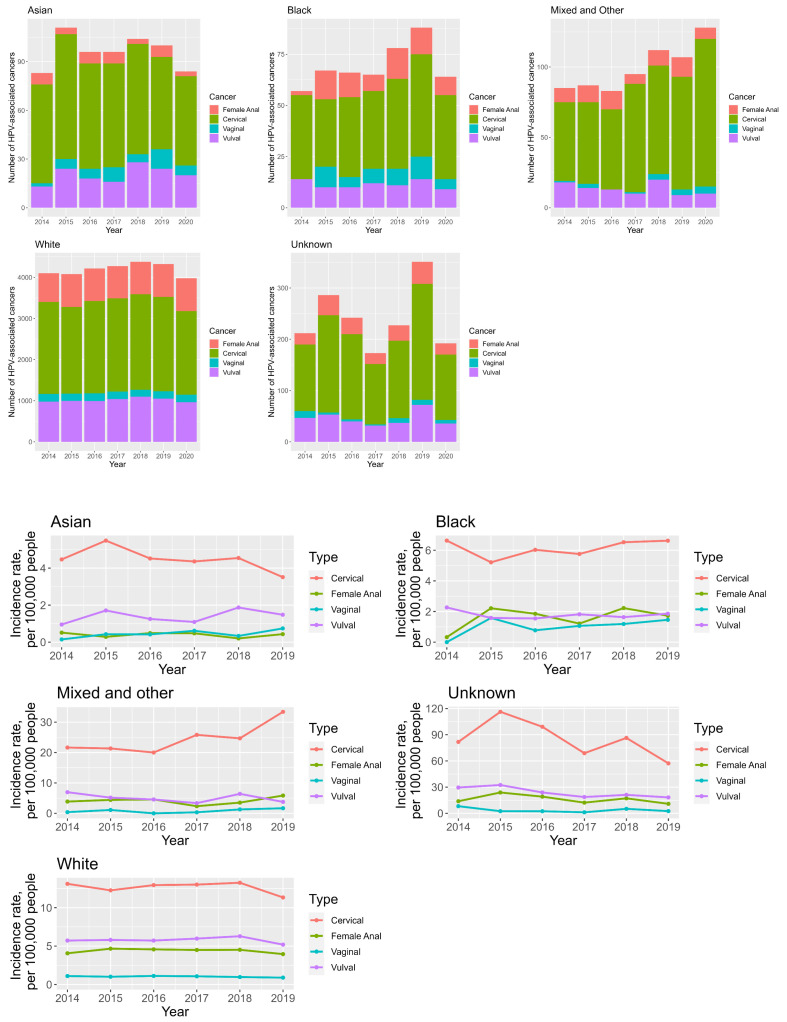
Stacked bar chart looking at the total numbers of anogenital cancers in women between 2014 and 2019, broken down in ethnicity groups (**top**). Line graphs demonstrating ethnicity—specific anogenital cancer incidence trends in women between 2014 and 2019 (**bottom**).

**Figure 4 cancers-16-02177-f004:**
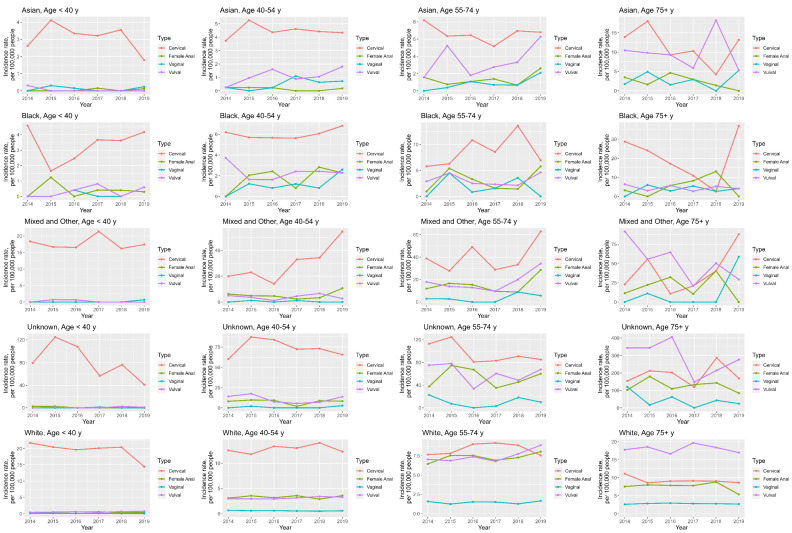
Line graphs demonstrating ethnicity and age—specific anogenital cancer incidence trends in women between 2014 and 2019.

**Figure 5 cancers-16-02177-f005:**
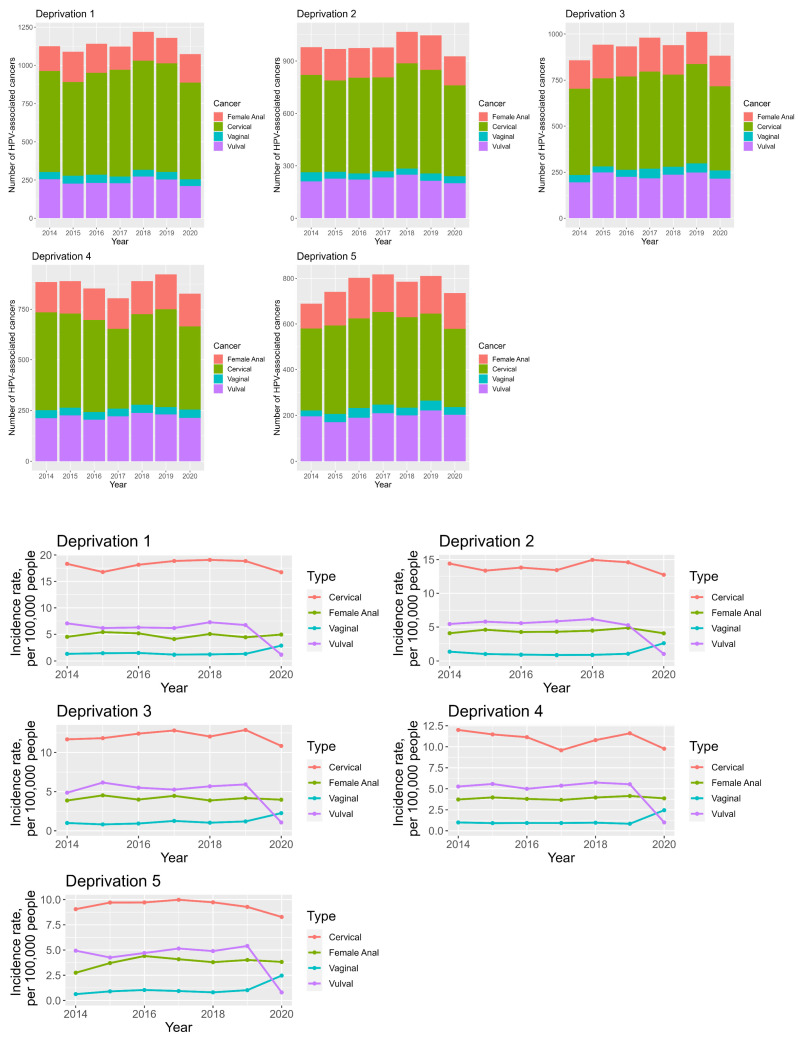
Stacked bar chart looking at the total numbers of anogenital cancers in women between 2014 and 2020, broken down in deprivation quintiles (**top**). Line graph demonstrating deprivation—specific anogenital cancer incidence trends in women between 2014 and 2020 (**bottom**).

**Figure 6 cancers-16-02177-f006:**
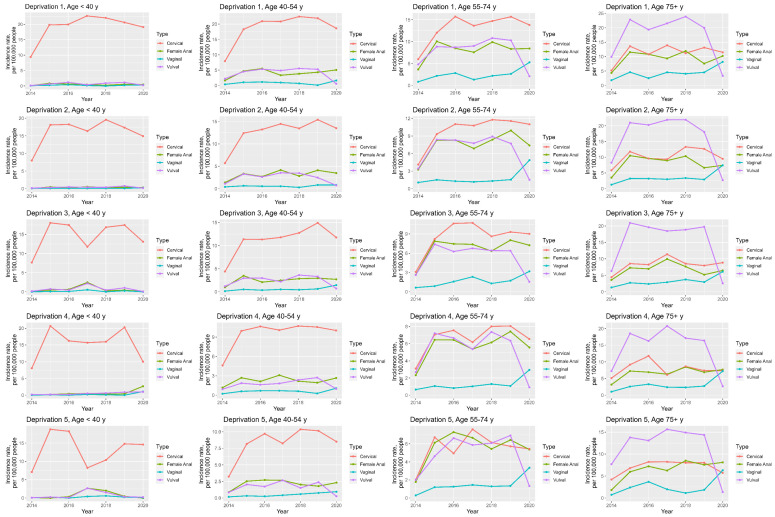
Line graphs demonstrating deprivation and age—specific anogenital cancer incidence trends in women between 2014 and 2020.

**Figure 7 cancers-16-02177-f007:**
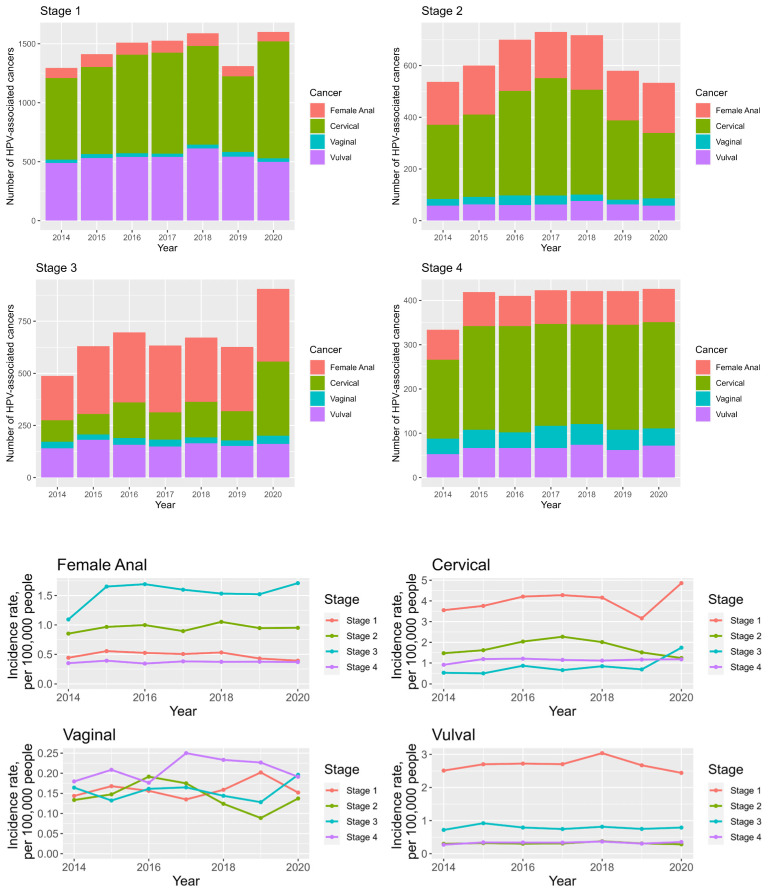
Stacked bar chart looking at the total numbers of anogenital cancers in women between 2014 and 2020, broken down by cancer staging (**top**). Line graph demonstrating anal, cervical, vulval and vaginal cancer incidence rates for staging between 2014 and 2020 (**bottom**).

**Figure 8 cancers-16-02177-f008:**
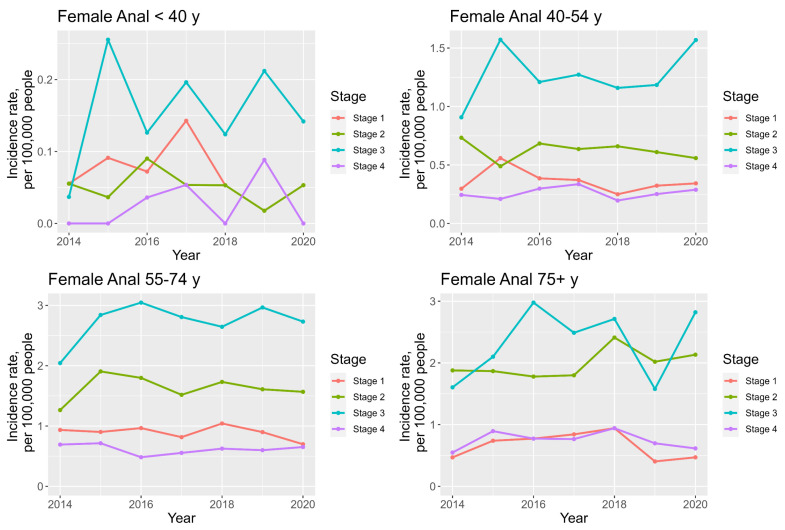
Line charts demonstrating staging and age—specific female anal cancer incidence trends between 2014 and 2020.

**Figure 9 cancers-16-02177-f009:**
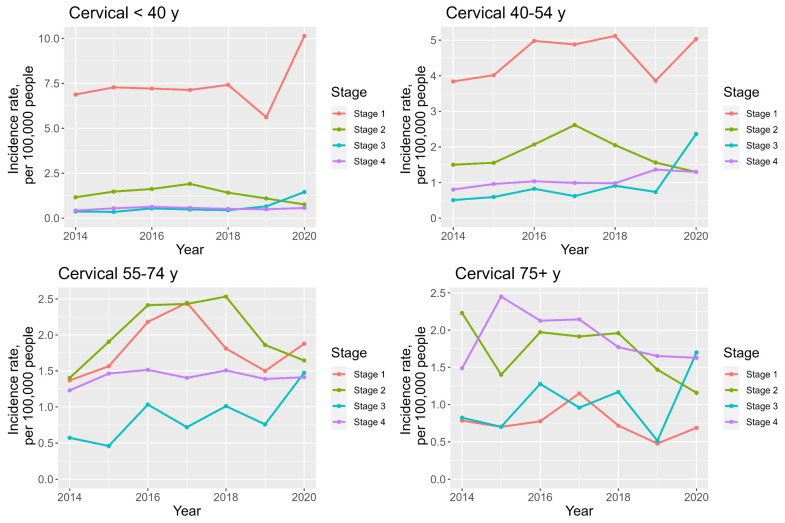
Line charts demonstrating staging and age—specific cervical cancer incidence trends between 2014 and 2020.

**Figure 10 cancers-16-02177-f010:**
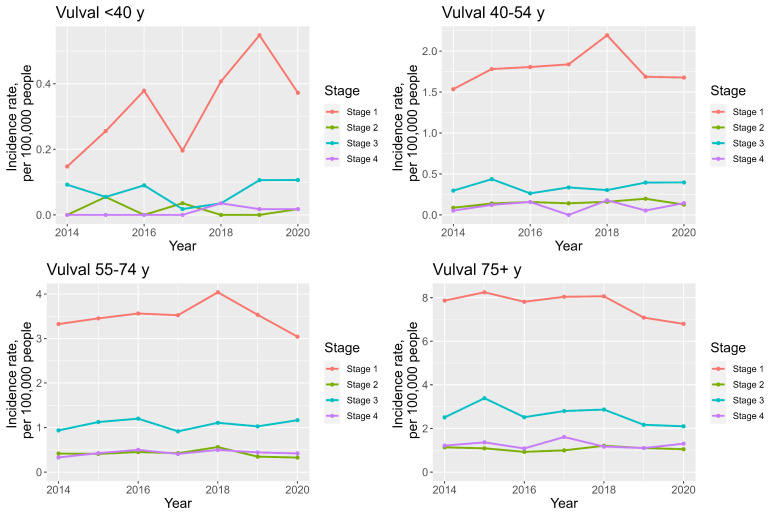
Line charts demonstrating staging and age—specific vulval cancer trends between 2014 and 2020.

## Data Availability

The complete datasets generated and analysed in this study are available to healthcare workers from the Cancer Outcomes Services Dataset maintained by NCRAS.

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
