# Peer review of "Anogenital HPV-Related Cancers in Women: Investigating Trends and Sociodemographic Risk Factors"

_cancers, 2024, doi:10.3390/cancers16122177_

Round 1

Reviewer 1 Report (Previous Reviewer 1)

Comments and Suggestions for Authors

The authors have addressed all my concerns and I am satisfied with the changes. This has resulted in a large number of edits and some revisions to the conclusions.

Reviewer 2 Report (Previous Reviewer 2)

Comments and Suggestions for Authors

I am happy with the review and have no further comments or questions.

Reviewer 3 Report (Previous Reviewer 3)

Comments and Suggestions for Authors

Dear Authors,

Thank you so much for giving me this opportunity, the manuscript is very well revised and easy to understand.

This manuscript is a resubmission of an earlier submission. The following is a list of the peer review reports and author responses from that submission.

Round 1

Reviewer 1 Report

Comments and Suggestions for Authors

Lupi et al use the CancerStat2 platform and associated demographic data to investigate the incidence of HPV associated anogenital cancers in England from 2014-2020 and determine if a connection exists between prior cervical cancer detection and later detection of other HPV related cancers.  I think this could be a useful and reasonably important study, but I suspect that some of the data analysis is flawed. 

Major Points:

1) The description of how changes in disease incidence over the 2014-2020 time period is not described in a way that I could interpret based on lines 138-140.  This must be explained more clearly. I think their methodology is flawed based on what is stated regarding calculating differences in quartiles.  Dividing incidence data in quartiles and using the upper and lower quartiles to calculate change would report the largest changes between any years, not across the full range of 2014 to 2020.  I suspect that creating a line of best fit across the time range and determining its slope would be more useful.  Commonly, other papers report changes in the form of "a 2.1% decrease in adjusted incidence per year across the time frame of 2014-2020 was observed".  Alternatively, perhaps calculating a change between each consecutive year and averaging these annual changes across the entire time frame might yield equivalent data more directly, providing an average annual change across the time frame in question.  I am not an expert in this area, and I advise the authors to consult with experts about more appropriate ways to calculate these value.  This is really important, as they comprise the vast majority of the presented data in the manuscript.

2) What is known about HPV vaccination status in each specific individual, in this cohort in general or even regarding vaccine availability or uptake in this population in general?  I would think vaccination data for each person should be in the database? Many countries are now reporting significant decreases in HPV related cervical cancers in younger women that are attributed to regional or national vaccine programs (such as PMID: 35294540 for example).  This is certainly something that deserves investigation or discussion at a minimum. Certainly, an overall trend in reduction in cervical cancer in younger women that would have had access to vaccine should be expected.

Minor Points:

1) line 42 - there are over 400 HPV types (PMID: 28053164), while they say over 100 types

2) line 49 - HPV does not cause 100% of cervical cancers.  Its more like 90% (PMID: 36497170).

3) The reduced number of cases of anogenital cancers detected in England in 2020 is quite likely an artifact of the COVID-19 pandemic, as most developed countries experienced reduced cancer screening during the pandemic (see things like PMID: 35965514). This may explain the dip in 2020 in figure 1 (top).

4) Some of the paragraph breaks are not needed.

Author Response

Major Points:

1) The description of how changes in disease incidence over the 2014-2020 time period is not described in a way that I could interpret based on lines 138-140.  This must be explained more clearly. I think their methodology is flawed based on what is stated regarding calculating differences in quartiles.  Dividing incidence data in quartiles and using the upper and lower quartiles to calculate change would report the largest changes between any years, not across the full range of 2014 to 2020.  I suspect that creating a line of best fit across the time range and determining its slope would be more useful.  Commonly, other papers report changes in the form of "a 2.1% decrease in adjusted incidence per year across the time frame of 2014-2020 was observed".  Alternatively, perhaps calculating a change between each consecutive year and averaging these annual changes across the entire time frame might yield equivalent data more directly, providing an average annual change across the time frame in question.  I am not an expert in this area, and I advise the authors to consult with experts about more appropriate ways to calculate these value.  This is really important, as they comprise the vast majority of the presented data in the manuscript.

Thank for your comment. We have re-consulted our statistician who has reviewed this. To investigate incidence trends between 2014 and 2020, we have now calculated the Average Annual Percentage Change (AAPC). This involved first calculating the annual percentage change in incidence between each consecutive year using the formula:

((Incidence Later year – Incidence Earlier year)/ Incidence Earlier year)*100

The AAPC was then obtained by averaging these annual percentage changes. The mean and standard deviation were selected for this and are presented in the tables in supplementary materials. Of note, it was not possible to carry out any statistical analysis on these trends. Multiple regression models including linear and joinpoint regression were trialled, however due to the small number of data points, these methods were not feasible

This has been explained in both the methods and in the ‘study limitations’ of the paper.

2) What is known about HPV vaccination status in each specific individual, in this cohort in general or even regarding vaccine availability or uptake in this population in general?  I would think vaccination data for each person should be in the database? Many countries are now reporting significant decreases in HPV related cervical cancers in younger women that are attributed to regional or national vaccine programs (such as PMID: 35294540 for example).  This is certainly something that deserves investigation or discussion at a minimum. Certainly, an overall trend in reduction in cervical cancer in younger women that would have had access to vaccine should be expected.

Thank you for your comments and for highlighting the importance of vaccination. We agree that vaccination is an extremely important topic, the introduction of the HPV immunisation program in England in 2008 has already reduced the risk of developing cervical HSIL by 97% in those patients vaccinated at 12-13 years, this is predicted to nearly eliminate cervical cancer in women born after 1/9/1995; as you correctly highlighted this is likely partly responsible for the continuous fall in cervical cancer incidence seen in women under 40 years of age,  as vaccinated women would be 25 years of age in 2020 (the upper range of our data collection). Although, it is also worth noting that our data is only just beginning to capture the potential benefit of vaccination in 2020 in a still very young patient cohort, which is only beginning to have cervical screening. Moreover, we are not going to be seeing it’s true impact on other anogenital cancers until at least 2050, when vaccinated women are expected to reach the peak age of vulval and anal cancer. It is also not benefitting women over the age of 55 years in who we are seeing a second peak of cervical cancer. In view of your suggestions, we have emphasised this in lines:  985-1050.
With respect to including vaccination data in the analysis, unfortunately this is not possible as COSD does not supply this information, the COSD data is simply not linked with immunisation records. We have made a note of this in the limitations of the study section.

Minor Points:

1) line 42 - there are over 400 HPV types (PMID: 28053164), while they say over 100 types

Thank you for the comment. I have amended this as recommended.

2) line 49 - HPV does not cause 100% of cervical cancers.  Its more like 90% (PMID: 36497170).

Thank you for the comment. I have amended this as recommended.

3) The reduced number of cases of anogenital cancers detected in England in 2020 is quite likely an artifact of the COVID-19 pandemic, as most developed countries experienced reduced cancer screening during the pandemic (see things like PMID: 35965514). This may explain the dip in 2020 in figure 1 (top).

Thank you for the additional reference. I have added this to the paper. We discuss the impact of the pandemic in lines 971-984.

4) Some of the paragraph breaks are not needed.

I have amended formatting accordingly.

In addition to the above, we have also reviewed our conclusion in view of your suggestions

Reviewer 2 Report

Comments and Suggestions for Authors

The aim of this study was to explore the incidence trends of anogenital cancers among women in England between 2014 and 2020, as well as studying possible risk factors such as age, deprivation, and ethnicity. Data from the Clinical Outcomes and Services Dataset was used for the study.

The main strength of the study is the access to a large dataset of anogenital cancers with additional data on demographics, ethnicity, and social deprivation.

To me, the main limitation of this study is the very limited statistical analyses being used which restricts the ability to answer the research questions of the study. More on that below.

Abstract:see comments under Discussion 

Introduction: This is a fair summary of the field with references that are up-to date. One could consider adding the following references that studies the prevalence of anal hrHPV in women with and without cervical hrHPV (Liu Y et al . J Infect Dis. 2023 Apr 18;227(8):932-938 and Wei et al. J Infect Dis. 2023 Feb 14;227(4):488-497)

Methods: A major part of the manuscript consists of incidence rates and the focus is on incidence trends. But there were no actual statistical tests performed to assess if the differences seen in trends were statistically different. Did you discuss with a statistician whether a statistical analysis could be performed to assess trends despite limited time points?

When studying potential factors associated with anogenital cancer only Chi2 or Fisher tests were used. This does not answer questions such as “was ethnicity associated with anogenital cancer” given that this could have been confounded by for example deprivation. I suggest using multivariable regression models (such as Poisson regression or Logistic regression depending on how you want to assess time in your models) to assess factors that may or may not be associated with the outcome. Possible interaction should also be assessed with appropriate statistical models.

Results: The result section is very long. If multivariable regression models are added, as suggested above, perhaps some Tables and Figures can be moved to Supplement.

Discussion:

·      If multivariable regression models are added I suggest that the Discussion (and Abstract) is updated regarding possible important factors associated with anogenital cancer.

·      I suggest adding a few lines in the discussion regarding how screening of women of high-risk could be done? By gynecologists who identify women at risk? Or by endoscopists? Other thoughts?

·      Under limitations I suggest adding that data regarding HIV status is missing given that this is an important risk factor.

Author Response

Introduction: This is a fair summary of the field with references that are up-to date. One could consider adding the following references that studies the prevalence of anal hrHPV in women with and without cervical hrHPV (Liu Y et al . J Infect Dis. 2023 Apr 18;227(8):932-938 and Wei et al. J Infect Dis. 2023 Feb 14;227(4):488-497)

 Thank you for the comment. We have integrated these references in the paper.  

Methods: A major part of the manuscript consists of incidence rates and the focus is on incidence trends. But there were no actual statistical tests performed to assess if the differences seen in trends were statistically different. Did you discuss with a statistician whether a statistical analysis could be performed to assess trends despite limited time points?

Thank for your comment. We have re-consulted our statistician who has reviewed this. To investigate incidence trends between 2014 and 2020, we have now calculated the Average Annual Percentage Change (AAPC). This involved first calculating the annual percentage change in incidence between each consecutive year using the formula:

 ((Incidence Later year – Incidence Earlier year)/ Incidence Earlier year)*100

The AAPC was then obtained by averaging these annual percentage changes. The mean and standard deviation were selected for this and are presented in the tables in supplementary materials. Of note, it was not possible to carry out any statistical analysis on these trends. Multiple regression models including linear and joinpoint regression were trialled, however due to the small number of data points, these methods were not feasible

This has been explained in both the methods and in the ‘study limitations’ of the paper.

 When studying potential factors associated with anogenital cancer only Chi2 or Fisher tests were used. This does not answer questions such as “was ethnicity associated with anogenital cancer” given that this could have been confounded by for example deprivation. I suggest using multivariable regression models (such as Poisson regression or Logistic regression depending on how you want to assess time in your models) to assess factors that may or may not be associated with the outcome. Possible interaction should also be assessed with appropriate statistical models.

 Thank you for your comments. I have re-consulted our statistician on the matter. We have now added additional multivariate logistic regression analysis to further assess the relationships between different variables. For the purpose of this analysis age was coded as below or above 55 years with age <55 years being set as the reference group. Deprivation was coded as high (quintiles 1-2) and low (quintiles 4-5) with low deprivation being set as the reference group. For ethnicity, white ethnicity was set as the reference group, and staging was coded as early (stage 1-2) and late (stage 3– 4) stage.  Adjusted odds ratios (ORs) for the outcome of interest were obtained by accounting for the presence of specific characteristics while controlling for the effects of other factors. The odd’s ratio (OR), 95% confidence interval (CI) and p-value are reported.

Results: The result section is very long. If multivariable regression models are added, as suggested above, perhaps some Tables and Figures can be moved to Supplement.

Thank you for your comment. We have created a supplementary document with all the tables.

Discussion:

  • If multivariable regression models are added I suggest that the Discussion (and Abstract) is updated regarding possible important factors associated with anogenital cancer.

Thank you for your comment. This has been amended accordingly.

  • I suggest adding a few lines in the discussion regarding how screening of women of high-risk could be done? By gynecologists who identify women at risk? Or by endoscopists? Other thoughts?

Thank you for your comment. We have previously discussed this in one of our papers ‘Lupi, Micol, et al. "Anal cancer in high-risk women: the lost tribe." Cancers 15.1 (2022): 60.’. Since then, the International Anal Neoplasia have published recommendations for anal cancer screening Stier, Elizabeth A., et al. "International Anal Neoplasia Society's consensus guidelines for anal cancer screening." International Journal of Cancer (2024). Whilst we agree that the logistics around the setting up of anal cancer screening is an important topic, it is also a complex topic limited by a general lack of research in high-risk women, resources, expertise and availability of high-resolution anoscopy. Given this we feel that this is not a topic which can be quickly added in the discussion and is beyond the scope of this paper. We have therefore decided not to include this in our paper. We hope you understand.

  • Under limitations I suggest adding that data regarding HIV status is missing given that this is an important risk factor.

Thank you for your comment, we have added this to the revised draft.

Reviewer 3 Report

Comments and Suggestions for Authors

I found the manuscript very interesting and valuable, and it is essential to note that anal cancer is increasing in women as an HPV-related disease and that HPV-infected individuals and those with prior HSIL are at higher risk of developing other HPV-related cancers.

I have several concerns.

1. did the authors not extract head and neck cancer data from the COSD database for this analysis? Although the title is ANOGENITALS, I would like to see it added. 

2. I wonder how many of the 32,893 cases of anogenital cancers had received HPV vaccination. 

I am wondering how many of the 32,893 anogenital cancers were vaccinated against HPV and what are the results of cervical cytology and HPV testing.

3. the definition of deprivation needs to be clarified. Literature is cited, but please describe the definition.

4. Lines 394-422 are a series of short sentences that are blurred in focus and difficult to understand.

5. Is the 2014-2019 data affected by the COVID-19 pandemic? Please consider how it is described around line 446 in the Discussion.

6. is the cervical cancer incidence rate in the UK going up as far as Figure 9 and others show? In England, are boys and girls aged 12-13 now vaccinated against HPV at school, and is the incidence rate declining? The vaccination rate is 86% for girls and 81% for boys.

7. Figures 5 and 6 could be more precise. Unless you make the graphs by organ, it is difficult to read the changes in the incidence of vaginal cancer. Is there any consideration as to why vulvar cancer is decreasing, and vaginal cancer is increasing?

Author Response

  1. did the authors not extract head and neck cancer data from the COSD database for this analysis? Although the title is ANOGENITALS, I would like to see it added. 

Thank you for your comment, we do not have the data for head and neck cancers. We wanted to focus on anogenital cancers in women given the proximity of the anus with other genital sites and the fact that women with genital driven HPV pathology have been shown to be at higher risk of anal cancer compared to the general population. These women therefore share characteristics which put them at higher risk of these conditions. 

  1. I wonder how many of the 32,893 cases of anogenital cancers had received HPV vaccination. 

Thank you for your comment. Vaccination is an extremely important topic, the introduction of the HPV immunisation program in England in 2008 has already reduced the risk of developing cervical HSIL by 97% in those patients vaccinated at 12-13 years, this is predicted to nearly eliminate cervical cancer in women born after 1/9/1995; this is likely partly responsible for the continuous fall in cervical cancer incidence seen in women under 40 years of age,  as vaccinated women would be 25 years of age in 2020 (the upper range of our data collection).
Nevertheless, it is also worth noting that our data is only just beginning to capture the potential benefit of vaccination in 2020 in a still very young patient cohort, which is only beginning to have cervical screening. Moreover, we are not going to be seeing it’s true impact on other anogenital cancers until at least 2050, when vaccinated women are expected to reach the peak age of vulval and anal cancer. It is also not benefitting women over the age of 55 years in who we are seeing a second peak of cervical cancer In view of your suggestions, we have emphasised this in lines:  985-1050.
With respect to including vaccination data in the analysis, unfortunately this was not possible as COSD does not supply this information, the COSD data is simply not linked with immunisation records. We have made a note of this in the limitations of the study section.

I am wondering how many of the 32,893 anogenital cancers were vaccinated against HPV and what are the results of cervical cytology and HPV testing.

Please see response above for HPV vaccination. Similarly to the above the COSD dataset does link cervical cytology and HPV testing to cancer status. The only way to link such data is to submit a data access release application (DARS) via NHS Digital, which carries a fee and requires ethics approval and is not freely available. We hope that the mASCARA database (Brogden, D. R. L., et al. "Improving outcomes for the treatment of anal squamous cell carcinoma and anal intraepithelial neoplasia." Techniques in Coloproctology 23 (2019): 1109-1111) which we launched in 2019 will help overcome some of the issues around data access. We have made a note of this in the limitations of the study section.

  1. the definition of deprivation needs to be clarified. Literature is cited, but please describe the definition.

We have added further depth to the definition of deprivation.

  1. Lines 394-422 are a series of short sentences that are blurred in focus and difficult to understand.

Thank you for highlighting this. I have re-phrased these paragraphs and hopefully they are better focused and easier to understand.

  1. Is the 2014-2019 data affected by the COVID-19 pandemic? Please consider how it is described around line 446 in the Discussion.

Thank you for your comments, we have revised this.

  1. is the cervical cancer incidence rate in the UK going up as far as Figure 9 and others show? In England, are boys and girls aged 12-13 now vaccinated against HPV at school, and is the incidence rate declining? The vaccination rate is 86% for girls and 81% for boys.

Figure 9 looks at staging and age specific incidence. It is not looking at the general incidence of cervical cancer overall. This is demonstrated in figure 1 (general incidence).  What it shows is the incidence of cervical cancer according to stage in each age group. I.e. in the group <40 and 40-5 years groups stage 1 disease is most prevalent and diagnosis of stage 1 disease at presentation has been increasing in these groups, whilst that of stage 2 disease decreasing.

With respect to vaccination please refer to the reply to your comment number 2.

  1. Figures 5 and 6 could be more precise. Unless you make the graphs by organ, it is difficult to read the changes in the incidence of vaginal cancer. Is there any consideration as to why vulvar cancer is decreasing, and vaginal cancer is increasing?

Thank you for your comments. Given the amount of data that these graphs are displaying we think this is the best way to display the patterns rather than crude numbers. You would need to refer to table 2 and 3 to look at the median incidences and interquartile range to view the values. We hope this is sufficient. With respect to your other comment, it is difficult to draw conclusions on the drop in vulval and rise in vaginal cancer seen in graph over the 2019-2020 time period; these are changes over 1 year only, moreover this is the year where COVID19 data was collected, which is probably misleading.  The release of more up to date data over the last 5 years is needed to understand the direction of change fully. Unfortunately, this data is still not available. We have included a sentence in the discussion and in the ‘study limitations’ section to address this uncertainty and limitations of the dataset.